# WIDE NEURAL NETWORKS ARE INTERPOLATING KERNEL METHODS: IMPACT OF INITIALIZATION ON GENERALIZATION

## ABSTRACT

The recently developed link between strongly overparametrized neural networks (NNs) and kernel methods has opened a new way to understand puzzling features of NNs, such as their convergence and generalization behaviors. In this paper, we make the bias of initialization on strongly overparametrized NNs under gradient descent explicit. We prove that fully-connected wide ReLU-NNs trained with squared loss are essentially a sum of two parts: The first is the minimum complexity solution of an interpolating kernel method, while the second contributes to the test error only and depends heavily on the initialization.

This decomposition has two consequences: (a) the second part becomes negligible in the regime of small initialization variance, which allows us to transfer generalization bounds from minimum complexity interpolating kernel methods to NNs; (b) in the opposite regime, the test error of wide NNs increases significantly with the initialization variance, while still interpolating the training data perfectly. Our work shows that – contrary to common belief – the initialization scheme has a strong effect on generalization performance, providing a novel criterion to identify good initialization strategies.

## 1 INTRODUCTION

Neural networks (NNs) have celebrated many successes over the past decade and achieved state of the art performance across various domains and tasks. From a theoretical standpoint, however, many aspects of neural networks are not well understood, as for example illustrated by Zhang et al. (2016). Neural networks seem to contradict classical learning theory as, in many scenarios, they are able to fit random labels perfectly while still generalizing well when trained on the true labels. In addition, overparametrized neural networks frequently exhibit even improved test performance when the number of parameters is increased further (Belkin et al., 2019). NN models thus often seem to avoid overfitting.

Very recently there have been advances in understanding the training and evaluation behavior of neural networks in the infinitely wide limit (Jacot et al., 2018; Hayou et al., 2019) and also in the strongly overparametrized regime (Du et al., 2018b; Li & Liang, 2018; Allen-Zhu et al., 2018a), which is close to the infinite limit NN. Both lines of work express the behavior of the neural network in terms of the so-called neural tangent kernel (NTK). In particular, Du et al. (2018b) showed in this way that, under mild conditions, strongly over-parametrized neural networks converge to a global minimum of zero training error.

In another line of research, Belkin et al. (2018b) suggested a new picture of learning in the overparametrized regime, introducing an "interpolating kernel method" which successfully learns in this regime. This method selects the least complex function that interpolates all data points perfectly, as opposed to traditional methods which balance the function's complexity against its goodness-of-fit. It is suggested that this picture of overparametrized learning could help understand neural networks.

If overparametrized NNs are indeed linked to such kernel methods, then zero (or small) training error alone might not tell us much about the test performance, as the measure of function complexity could be determined by the NN architecture, initialization, and the optimization method. A bad NN design may lead to an unfavorable complexity measure that could result in a large generalization gap.

The question therefore remains in which way exactly are overparametrized neural networks linked to interpolating kernel methods? And how can this connection help us better understand neural networks and their puzzling features?

## 1.1 OUR CONTRIBUTIONS

In this paper we answer the first question by making the link to interpolating kernel methods explicit for strongly overparametrized NNs. After that, we exhibit two implications for NNs which this connection allows us to draw.

First, for strongly overparametrized ReLU-NNs we make the bias explicit that is imposed by the NN initialization and by training with discrete gradient descent steps on the squared loss. We achieve this by decomposing the learned NN function as well as the test error into two terms: The first term corresponds to a minimum complexity interpolating kernel method, whereas the second term is proportional to the initialized weights and gives a non-zero contribution only if the test input is not part of the training set.

Second, we are able to bound the difference between a NN with standard initialization of small variance and the solution of a minimum complexity interpolating kernel method based on the NTK. This bound provides a new way to transfer existing test error bounds from interpolating kernel methods to NNs, and vice versa.

Our third contribution shows that without putting any constraints on the NN initialization, low training error does not imply low test error. This is because the second term in the above test error decomposition grows as a power of the initialization variance, as we show, whereas the first term is independent of this variance. Thus, the test error grows dramatically with increasing initialization variance, despite (close to) zero training error. Our findings provide an additional theoretical way to analyze initialization schemes, besides the existing heuristics and intention to prevent vanishing and exploding gradients (He et al., 2015; Glorot & Bengio, 2010).

Our theoretical bounds and insights are nicely corroborated in several experimental settings.

## 1.2 NOTATION

We consider a supervised learning setting with data $\mathcal{D} = \{(x_i, y_i)\}_{i=1}^N \subset B_1^d(0) \times \mathbb{R}$ consisting of $N$ training points, i.e. the inputs come from the bounded $d$-dimensional ball $B_1^d(0) = \{x \in \mathbb{R}^d : ||x||_2 \leq 1\}^1$. We often view the training inputs $X = (x_1, \ldots, x_N)^T \in \mathbb{R}^{N \times d}$ and corresponding labels $Y = (y_1, \ldots, y_N)^T \in \mathbb{R}^N$ in matrix form. For any function $g$ with domain $\mathbb{R}^d$, we define $g(X)$ as the row-wise application of $g$. Similarly, for any set $\mathcal{D}_{\text{test}} \subset \mathbb{R}^d \times \mathbb{R}$ of $N_{\text{test}}$ test points we define $X_{\text{test}} \in \mathbb{R}^{N_{\text{test}} \times d}$ and $Y_{\text{test}} \in \mathbb{R}^{N_{\text{test}}}$. We call a dataset $\mathcal{D}$ consistent if $x_i = x_j$ implies $y_i = y_j$.

We denote the output of a fully-connected NN with $L$ layers and parameters $\theta$ by

$$
\begin{aligned}
f_\theta^{\text{NN}}(x) &= \frac{\sigma}{\sqrt{m_L}} W^L a(h^{L-1}(x)) + \sigma b^L, \\
h^l(x) &= \frac{\sigma}{\sqrt{m_l}} W^l a(h^{l-1}) + \sigma b^l \quad \text{for } l = 2, \ldots, L-1, \\
h^1(x) &= \frac{\sigma}{\sqrt{m_1}} W^1 x + \sigma b^1 \in \mathbb{R}^d,
\end{aligned}
\tag{1}
$$

where $a(h) := \max\{h, 0\}$ is the ReLU-activation function applied componentwise, $m_l$ denotes the number of neurons in layer $l$, and $W^l \in \mathbb{R}^{m_l \times m_{l-1}}$ and $b^l \in \mathbb{R}^{m_l}$ are the weights and biases. For simplicity we choose the number of units in each hidden layer to be the same[2] $m_l = m$ for $l = 1, \ldots, L-1$ and $m_L = 1$. $\theta^l = [W^l, b^l]$ denotes the vector of weights and biases in layer $l$, and $\theta = [\theta^1, \ldots, \theta^L]$ the vector of all parameters, whose total number is $M = \sum_{l=1}^L (m_{l-1} + 1) m_l$. We keep the "initialization variance" $\sigma^2$ (equivalently, $\sigma > 0$) as an explicit scalar parameter, that can be varied.

---

[1] We consider one-dimensional outputs in this paper, although this is easy to generalize.

[2] Our arguments hold for general fully-connected NNs, too in the "wide limit" $\min_{l=1,\ldots,L-1}\{m_l\} \to \infty$.

Our parametrization (1) together with a standard normal initialization $W^l_{i,j}, b^l_i \sim \mathcal{N}(0,1)$ is the so-called NTK-parametrization (e.g. Du et al. (2018b)). In a forward pass this is equivalent to the standard parametrization (setting $\sigma/\sqrt{m_l} \to 1$ in Eq. (1)) with the weights and biases initialized at variances $\sigma^2/m_l$ and $\sigma^2$, respectively (He et al., 2015). During gradient training the two parametrizations are equivalent up to a parameter-dependent scaling factor of the learning rate (Lee et al., 2019).

We consider the squared error loss $\ell(\hat{y}, y) = |\hat{y} - y|^2/2$ and train by minimizing the empirical risk

$$L_{\mathcal{D}}(\theta) = \sum_{(x,y)\in\mathcal{D}} \ell(f^{\mathrm{NN}}_\theta(x), y) = \|f^{\mathrm{NN}}_\theta(X) - Y\|^2_2/2 \tag{2}$$

using gradient descent (GD). That is, starting from the initialization $\theta = \theta_0$ the weights are updated according to the discrete iteration $\theta_{t+1} = \theta_t - \eta\nabla_\theta L(\theta_t)$ for $t = 0, 1, \ldots$, where $\eta$ is a learning rate.

We connect NNs to interpolating kernel methods w.r.t. two specific kernel functions that are defined for any initialization $\theta = \theta_0$ of a NN (1). The first kernel $\Psi : \mathbb{R}^d \times \mathbb{R}^d \to \mathbb{R}$ is associated with training only the last layer $\theta^L$ of a NN (Lee et al., 2019),

$$\Psi(x, x') = \psi(x)\psi(x')^T, \quad \text{where} \;\; \psi(x) = \nabla_{\theta^L} f^{\mathrm{NN}}_\theta(x)\big|_{\theta=\theta_0} \in \mathbb{R}^{1\times(m+1)} \tag{3}$$

is the corresponding feature map. The second is the so-called neural tangent kernel (NTK) $\Phi : \mathbb{R}^d \times \mathbb{R}^d \to \mathbb{R}$ associated with training all NN parameters (Jacot et al., 2018; Lee et al., 2019),

$$\Phi(x, x') = \phi(x)\phi(x')^T, \quad \text{where} \;\; \phi(x) = \nabla_\theta f^{\mathrm{NN}}_\theta(x)\big|_{\theta=\theta_0} \in \mathbb{R}^{1\times M} \tag{4}$$

again represents the feature map. The "minimum complexity interpolating kernel method" fits a consistent training set $\mathcal{D}$ perfectly, using a function of minimum kernel norm w.r.t. a kernel $K$:

$$f^{\mathrm{int}}_K := \arg\min_{f\in\mathcal{H}_K} \|f\|_{\mathcal{H}_K} \quad \text{subject to} \;\; Y = f(X), \tag{5}$$

where $\mathcal{H}_K$ is the reproducing kernel Hilbert space (RKHS) associated with $K$ (Belkin et al. (2018b); see also App. B). Additionally, we denote the solution of a fully converged fully-connected ReLU-NN trained with gradient descent, squared loss and random initialization by $f^{\mathrm{NN}}$. We denote the test losses for these two predictors by $L^{\mathrm{meth}}_{\mathrm{test}} = \|f^{\mathrm{meth}}(X_{\mathrm{test}}) - Y_{\mathrm{test}}\|^2/(N_{\mathrm{test}})$ for meth = NN, int.

## 2 RELATED WORK

There are two main lines of research related to our work. The first line investigated in which way strongly overparametrized NNs, where the number $m$ of units per hidden layer scales polynomially with the number $N$ of training data points, converge to arbitrary small training error during training. The first papers to investigate this for 1 hidden layer were Daniely (2017), Li & Liang (2018) and Du et al. (2018b). Later, Allen-Zhu et al. (2018a), Allen-Zhu et al. (2018b), Zou et al. (2018) and Du et al. (2018a) extended the work to deeper networks, CNNs and RNNs (Goodfellow et al., 2016). Li & Liang (2018) mostly focus on the cross entropy loss, whereas Du et al. (2018b) focuses on the squared error as we do here. Additionally, Du et al. (2018b) is the first work to connect the training behaviour of finite NNs (not only in the limit of infinite width $m$) to the NTK-kernel introduced by Jacot et al. (2018). This connection is important for our work. In a similar vein, Arora et al. (2019b) developed data dependent generalization bounds based on the fact that the weights do not move far from their initialization, as shown in the above works. Furthermore, Arora et al. (2019a) experimentally evaluate the generalization behaviour of the infinite width solution of NNs in the NTK-limit.

Most of these contributions focus on showing that the training loss converges to a global minimum or that the training dynamics is close to a trajectory that converges to zero training loss. We build upon these works, but in contrast our focus is the connection between the test error of a trained NN and the error of an interpolating kernel method. We achieve this by resolving the implicit bias that the initialization and gradient descent (GD) has on the test loss. The initialization bias on the training loss is minor compared to the test loss. The most related work to ours is by Zhang et al. (2019). They investigate a similar issue as we do but use a different method to weaken the effects

of non-zero initialization and also do not quantify the impact of initialization on generalization but give evidence for such an effect. The recent contribution Lee et al. (2019), building upon Lee et al. (2017), Matthews et al. (2018) and Neal (1996), explicitly solves the training dynamics and shows that the solution in the trained infinite model seems to be similar to a Gaussian Process, even though its covariance is not the Bayesian posterior of a simple prior. The solution of their continuous training time analysis is similar to the linearized model in the proof of our Thm. 2. We perform a discrete-time analysis, as in actual NN training, instead of the continuous-time (ODE-like) analysis of other works. Woodworth et al. (2019) uses a continuous time analysis as well to focus on the effect of noise in the gradients. Borovykh (2019) investigates the effect of the scaling between the "deep" and the "kernel" regime.

The second line of related work introduces interpolating kernel methods and the idea of an over-parametrized learning regime which does not suffer from overfitting even though the number of parameters is be much larger than the number of training samples. This line of work was mainly introduced by Belkin et al. (2018b) and the follow-up works Ma et al. (2017); Belkin et al. (2018c;a). The main idea is, instead of regularizing the function while trying to fit the data points well, to regularize (the complexity of) a function that perfectly interpolates all data points. This newly identified overparametrized learning regime challenges the conventional thinking about the bias-variance-tradeoff (Vapnik, 2013) from the underparametrized regime. Belkin et al. (2019) introduces a bouble-dip picture to describe the transition between the two regimes. Liang & Rakhlin (2018) have derived generalization bounds for these interpolating kernel methods using novel techniques. Using our explicit link to NNs, this will provide an alternative to NN generalization bounds like margin or PAC-Bayes-based bounds (e.g., Bartlett et al. (2017); Neyshabur et al. (2017)). While a connection between overparametrized NNs and interpolating kernel methods has been suggested in the papers above, the link has not been made explicit in a rigorous way, to the best of our knowledge.

The implicit bias of (stochastic) gradient descent ((S)GD) and random initialization on the trained NN has been the topic of many previous works, e.g. Soudry et al. (2018); Glorot & Bengio (2010); Daniely et al. (2016); Rahaman et al. (2018); Oymak & Soltanolkotabi (2018), which all elucidate certain aspects of the implicit bias. Our work is unique in that it investigates the effect (implicit bias of (S)GD and random initialization) on the test error of trained NNs, exhibiting NNs where – despite always vanishing training error – the test error can be either very good or arbitrarily large. Without gradient descent training and in a deterministic fashion, such examples have been constructed in (Mücke & Steinwart, 2019).

## 3 LINKING WIDE NNS AND INTERPOLATING KERNEL METHODS

We now develop the link between strongly overparametrized neural networks and minimum complexity interpolating kernel methods mathematically. We first illustrate the main idea with the simpler case where only the last NN layer is trained, before extending the result to general NN training.

### 3.1 PARADIGMATIC LINEAR CASE: TRAINING ONLY THE LAST NN LAYER

As the NN function $f_\theta^{\text{NN}}(x)$ (1) is linear in the parameters $\theta^L$ of the last layer, the associated features $\psi(x) = \nabla_{\theta^L} f_\theta^{\text{NN}}(x)\big|_{\theta_0}$ and corresponding kernel $\Psi$ from (3) do not depend on the last-layer initialization $\theta_0^L$ (but depend on the other layers $\theta_0^{1:L-1}$), and $f_\theta^{\text{NN}}(x) = \langle \theta^L, \psi(x) \rangle$ holds exactly for all $\theta$ and $x$. Starting NN gradient descent at $\theta_0^L = 0$, we get the following simple link:

**Theorem 1** *Let $f^{\text{NN}}$ be the fully converged solution of a fully-connected strongly overparametrized ReLU-NN (1) with $L$ layers, where only the last layer $\theta^L = [W^L, b^L]$ has been trained, using gradient descent under squared loss on a consistent dataset $\mathcal{D}$, after it had been initialized at $\theta_0^L = 0$, and all other layers have been kept at their random initialization $\sim \mathcal{N}(0, 1)$. Then it holds that*

$$f^{\text{NN}}(x^*) = f_\Psi^{\text{int}}(x^*) \qquad \forall x^* \in \mathbb{R}^d, \tag{6}$$

*where $f_\Psi^{\text{int}}$ is the minimum complexity interpolating kernel solution (5) of $\mathcal{D}$ w.r.t. kernel $\Psi$ (3).*

**Proof.** Abbreviating the last-layer parameters by $\vartheta = \theta^L$ and using the aforementioned linearity $f_\theta(x) = \langle \vartheta, \psi(x) \rangle$ in $\vartheta$, the gradient descent update rule reads:

$$\vartheta_{t+1} = \vartheta_t - \eta \psi(X)^T (\psi(X) \vartheta_t - Y). \tag{7}$$

This iteration can be solved explicitly using induction and the binomial theorem (Shah et al., 2018):

$$
\begin{aligned}
\vartheta_t &= \vartheta_0 + \psi(X)^T \left( \sum_{i=1}^{t} (-1)^{i-1} \binom{t}{i} \eta^i (\psi(X)\psi(X)^T)^{i-1} \right) (Y - \psi(X)\vartheta_0) \\
&= \vartheta_0 + \psi(X)^T (\psi(X)\psi(X)^T)^{-1} \left[ \mathbb{1} - \left( \mathbb{1} - \eta\psi(X)\psi(X)^T \right)^t \right] (Y - \psi(X)\vartheta_0),
\end{aligned}
\tag{8}
$$

where $(\psi(X)\psi(X)^T)^{-1}$ denotes the pseudoinverse. If the learning rate satisfies $\eta < 2/\lambda_{\max}(\psi(X)\psi(X)^T)$ such that gradient descent converges, we get $\left( \mathbb{1} - \eta\psi(X)\psi(X)^T \right)^t \to 0$ in the limit $t \to \infty$ as the spectral radius is smaller than one and the term in square brackets can be simplified to $\mathbb{1}$. To be precise this holds since $\mathcal{D}$ is consistent and we are in the strongly over-parametrized regime, so that $Y$ (and $\psi(X)\vartheta_0$) lies in the range of $\psi(X)\psi(X)^T$ almost surely (over the random initialization $\theta_0^{1:L-1}$, on which $\psi$ depends), see also Du et al. (2018b).

Setting $\vartheta_0 = 0$ as presupposed, we obtain the following predictor at full convergence ($t = \infty$):

$$
f^{\mathrm{NN}}(x^*) = \langle \vartheta_\infty, \psi(x^*) \rangle = \psi(x^*)\psi(X)^T(\psi(X)\psi(X)^T)^{-1}Y.
\tag{9}
$$

The same value $\vartheta_\infty$ can be obtained as the solution to the following least squares interpolation:

$$
\vartheta_\infty = \arg\min_\vartheta \|\vartheta\|_2^2 \quad \text{subject to } Y = \psi(X)\vartheta.
\tag{10}
$$

This can in turn be written as minimizing the RKHS norm of an interpolation function w.r.t. the kernel $\Psi$ from Eq. (3) (for details on this see App. A and B):

$$
f_\Psi^{\mathrm{int}} = \arg\min_{f \in \mathcal{H}_\Psi} \|f\|_{\mathcal{H}_\Psi} \quad \text{subject to } Y = f(X).
\tag{11}
$$

The claim has thus been proven by deriving the explicit solution (9) and showing it to equal $f_\Psi^{\mathrm{int}}$. $\square$

## 3.2 General case: Training the full Neural Network

In the case of training the full NN, we want to follow similar steps as in Sect. 3.1. For this, we first approximate $f_\theta^{\mathrm{NN}}(x)$ affine-linearly by $f_\theta^{\mathrm{lin}}$ around its initialization $\theta = \theta_0$, using recent results on wide NNs (Du et al., 2018b). We further drop the requirement that any components of $\theta_0$ must vanish; note, e.g., that the suggestive requirement $\theta_0 = 0$ (analogous to Sect. 3.1) would, for $L \geq 1$, lead to vanishing features $\phi(x) = \nabla_\theta f_\theta^{\mathrm{NN}}(x)\big|_{\theta_0}$ and a degenerate kernel $\Phi = 0$. Nonzero $\theta_0$ will however result in an initialization bias in $f_\theta^{\mathrm{lin}}$ as we will see (Eq. (14)).

**Theorem 2** *Let $f^{\mathrm{NN}}$ be the fully converged solution of a strongly overparametrized ReLU-NN (1) with $L$ layers, each with $m$ neurons, where all layers $\theta$ have been trained using gradient descent under squared loss on a consistent dataset $\mathcal{D}$ after random initialization $\theta_0 \sim \mathcal{N}(0, 1)$. Then there exists a $N \in \mathbb{N}$ such that for all NN with $m \geq N$ and $\forall x^* \in B_1^d(0)$ it holds with probability at least $1 - 2\delta$ over the initialization that*

$$
|f^{\mathrm{NN}}(x^*) - f_\Phi^{\mathrm{int}}(x^*)| \leq \frac{\sigma^L}{\delta^{1/2}} + O\left(\frac{1}{m^{1/2}}\right)
\tag{12}
$$

*where $f_\Phi^{\mathrm{int}}$ is the minimum complexity interpolating kernel solution (5) of $\mathcal{D}$ w.r.t. kernel $\Phi$ (4).*

For the proof of Thm. 2 we describe here especially those steps which are new compared to Thm. 1. The first property we need is the approximate linearity of the NN in its parameters $\theta$ during the whole gradient training. This is the reason why we restrict to strongly overparametrized fully-connected NNs. For such networks, a combination of the works (Jacot et al. (2018); Du et al. (2018b); Lee et al. (2019)) shows that the solution $\theta_t$ of the training dynamics $t = 0, 1, \ldots$ stays close to the initialization $\theta_0$, and the deviation from a linear model vanishes with growing width $m$:

**Lemma 1 (Lee et al. (2019); Du et al. (2018b))** *Denote the linearization of an NN (1) around its initialization $\theta_0$ by $f_\theta^{\mathrm{lin}}(x) := f_{\theta_0}^{\mathrm{NN}}(x) + (\nabla_\theta f_\theta^{\mathrm{NN}}(x)\big|_{\theta_0})(\theta - \theta_0)$. Further, for $t \geq 0$, let $\tilde{\theta}_t$ and $\theta_t$ be the parameters obtained by gradient descent training starting from $\theta_0$ with sufficiently small step size $\eta$ of the full NN and its linearisation respectively. Then, there exists some $N \in \mathbb{N}$ such that for all $m \geq N$ it holds with probability at least $1 - \delta$ over the initialization that: $\sup_t |f_{\tilde{\theta}_t}^{\mathrm{NN}}(x) - f_{\theta_t}^{\mathrm{lin}}(x)|^2 \leq O(1/m)$ for all $x \in B_1^d(0)$.*

Similar linearisation hold for strongly overparametrized RNNs and CNNs as well, not only for fully-connected NNs. The theorem proves that for every sufficiently wide NN the linearised solution is close to the trained full NN. For a more detailed analysis of the dependence of $N$ on the width $m$, variance $\sigma$ and $\delta$ we refer to Lee et al. (2019); Du et al. (2018b). Using this result, we continue our analysis with $f^{\text{lin}}$ which we can rewrite as $f_{\theta}^{\text{lin}}(x) = f_{\theta_0}^{\text{NN}}(x) + \langle \theta - \theta_0, \phi(x) \rangle = \left( f_{\theta_0}^{\text{NN}}(x) - \langle \theta_0, \phi(x) \rangle \right) + \langle \theta, \phi(x) \rangle$ were $\phi(x) = \nabla_\theta f_\theta^{\text{NN}}(x)|_{\theta_0}$ is the feature vector defined in Eq. (4). Similar to the proof of Thm. 1, gradient descent on $f_\theta^{\text{lin}}$ with the loss function $\|f_\theta^{\text{lin}}(X) - Y\|^2/2$ and sufficiently small step size $\eta < 2/\lambda_{\max}(\psi(X)\psi(X)^T)$ leads to the final ($t \to \infty$) parameter $\theta_\infty^{\text{lin}} = \theta_0 + \phi(X)^T \left( \phi(X)\phi(X)^T \right)^{-1} (Y - f_{\theta_0}^{\text{NN}}(X))$, resulting in the function

$$f^{\text{lin}}(x) = f_{\theta_0}^{\text{NN}}(x) + \phi(x)\phi(X)^T(\phi(X)\phi(X)^T)^{-1}(Y - f_{\theta_0}^{\text{NN}}(X)). \tag{13}$$

To connect Eq. (13), and thereby $f^{\text{NN}}$, to the solution $f_\Phi^{\text{int}}$ of the interpolating model, two steps remain to be done. First we show that the two summands in Eq. (13) which are not proportional to the labels $Y$ can be simplified into a single form for fully-connected ReLU-NNs; this can be done by showing $f_{\theta_0}(x) = \frac{1}{L}\langle \theta_0, \Phi(x) \rangle$ (see Lemma 2 for more details). Secondly, we can bound this single expression from above in terms of the initialization variance (Lemma 3; for proofs see Appendix. The term proportional to $Y$ is then identified with $f^{\text{int}}$.

**Lemma 2** *For a fully-connected ReLU-NN $f_\theta^{\text{NN}}$ (Eq. 1) with $L$ layers, the trained linearized version $f^{\text{lin}}$ from Eq. (13) can be written as follows:*

$$f^{\text{lin}}(x) = \phi(x)\phi(X)^T(\phi(X)\phi(X)^T)^{-1}Y + \frac{1}{L}\phi(x)\left[\mathbb{1} - \phi(X)^T(\phi(X)\phi(X)^T)^{-1}\phi(X)\right]\theta_0. \tag{14}$$

*Thus, the trained $f^{\text{NN}}(x) = f^{\text{lin}}(x) + O(1/\sqrt{m})$ decomposes essentially into the two terms of Eq. (14) (for $x \in B_1^d(0)$).*

As $|f^{\text{NN}}(x) - f^{\text{lin}}(x)| \le O(1/\sqrt{m})$, the expression (14) holds for the trained $f^{\text{NN}}$ in the wide limit $m \to \infty$ as well. This technical result enables all of our results in Thms. 2, 3, and 4.

Eq. (14) can be understood intuitively (in the case of a consistent $\mathcal{D}$): The first term makes the model interpolate the training data $\mathcal{D}$ perfectly ($f^{\text{lin}}(X) = Y$), independently of the initialization $\theta_0$, whereas the second term vanishes on the training data $x = X$ and thus only contributes to test inputs $x = x^*$. The second term furthermore depends on the initialization $\theta_0$, more precisely on the component of $\theta_0$ that is orthogonal to the feature manifold spanned by the data $\mathcal{D}$, i.e. orthogonal to the range of $\phi(X)^T$. Note that $\left[\mathbb{1} - \phi(X)^T(\phi(X)\phi(X)^T)^{-1}\phi(X)\right]$ is the projector onto the kernel of $\phi(X)$, about which the dataset $\mathcal{D}$ does not give any information.

**Lemma 3** *For a fully-connected ReLU-NN $f_\theta^{\text{NN}}$, it holds for all $x \in B_1^d(0)$ with probability at least $1 - \delta$ over the initialization $\theta_0$ that $\frac{1}{L}|\phi(x)\left(\mathbb{1} - \phi(X)^T(\phi(X)\phi(X)^T)^{-1}\phi(X)\right)\theta_0| \le \frac{\sigma^L}{\sqrt{\delta}}$.*

As in the proof of Thm. 1 (see Eqs. (9)–(11)) the first part of the decomposition in Eq. (14) is $f_\Phi^{\text{int}}$. By Lemma 3, we see that for small $\sigma^2$ the distance between $f^{\text{lin}}$ and $f_\Phi^{\text{int}}$ becomes small with high probability, and $f^{\text{lin}}$ and $f^{\text{NN}}$ are close too because of $m \gg N$. The triangle inequality $|f^{\text{NN}}(x^*) - f_\Phi^{\text{int}}(x^*)| \le |f^{\text{NN}}(x^*) - f_{/Phi}^{\text{lin}}(x^*)| + |f^{\text{lin}}(x^*) - f^{\text{int}}(x^*)|$ gives the desired result. Thus, strongly overparametrized neural networks initialized with small variance and trained under gradient descent correspond to solutions of a minimum complexity interpolating kernel method. $\square$

## 4 IMPLICATIONS FOR NEURAL NETWORKS

Now that we have established the link between NNs and interpolating kernel methods, we show two implications this has for the test loss of neural networks. First, we derive an upper bound on the NN test loss, which – in the regime of small NN initialization variance – allows us to transfer generalization bounds from interpolating kernel methods to neural networks. Second, a lower bound on the NN test loss gives – in the regime of large NN initialization variance – examples of NNs, trained under gradient descent after random initialization, that generalize badly despite (close to) zero training loss.

## 4.1 Transferring Generalization Bounds to Neural Networks

From Thm. 2, it is now almost straightforward to connect the test error of the interpolating kernel method to the test error (w.r.t. squared loss) of the neural network.

**Theorem 3** *Let $f^{\text{NN}}$ be the fully-converged solution of a strongly overparametrized NN (1) with $L$ layers and $m$ hidden units per layer and initialization variance $\sigma^2$, where all layers $\theta$ have been trained using gradient descent under squared loss on a consistent dataset $\mathcal{D}$ after random initialization $\theta_0 \sim \mathcal{N}(0,1)$. Then it holds with probability at least $1 - 2\delta$ over the random initialization that $\sqrt{L_{\text{test}}^{\text{NN}}} \leq \sqrt{L_{\text{test}}^{\text{int}}} + \sigma^L/\sqrt{\delta} + O(1/\sqrt{m})$.*

To prove this theorem we use the fact that $|f^{\text{lin}}(x) - f^{\text{NN}}(x)| \leq O(1/\sqrt{m})$ holds uniformly for all $x \in B_1^d(0)$ and that $\|f^{\text{lin}}(X_{\text{test}}) - f^{\text{int}}(X_{\text{test}})\|^2/N_{\text{test}} \leq \sigma^{2L}/\delta$ holds with high probability; the last fact follows in the same way as in the proof of Lemma 3 by, instead of applying Markov's inequality to the squared deviation for a single test point $x$, applying it to averaged sum of squared deviations over all test points. Then, we make use of the triangle inequality and evaluate the square to prove the desired result. This inequality now enables us to transfer bounds for the test error from one formalism to the other and gives us a new grip on finding bounds on the test error for NNs. Good generalization bounds on minimum complexity interpolating models have been established in (Liang & Rakhlin (2018)). On the other hand, Belkin et al. (2018b) showed that conventional bounds in the underparametrized regime might be of little use for minimum complexity kernel methods.

## 4.2 Zero NN training error alone does not imply small test error

In the past many people trained neural networks with the goal to achieve a low training error with the hope that this might also result in a low test error. The initialization was merely thought of as a necessity to converge faster to a low training error or to give a favourable starting point for the training dynamics. We now show with our analysis that random initialization alone is not enough, but instead we need certain constraints on the initialization to be able to give generalization bounds.

Before we prove this let us derive some simple properties. Let $\Phi(x, \sigma)$ be the feature vector of a NN initialized with NTK-parametrization with initialization variance $\sigma$. Then it is easy to see that $\Phi(x, \sigma) = \sigma^L \Phi(x, 1)$ because of the homogeneity of the derivative w.r.t constants. Further we define $J(X_{\text{test}}, \sigma) := \|\frac{1}{L}\Phi(X_{\text{test}}) \left(\mathbb{1} - \Phi(X)^T(\Phi(X)\Phi(X)^T)^{-1}\Phi(X)\right)\theta_0\|_2/\sqrt{N_{\text{test}}}$ and $J(X_{\text{test}}) := J(X_{\text{test}}, 1)$. With this notation, we can show the following theorem:

**Theorem 4** *Let $f^{\text{NN}}$ be the fully-converged solution of a strongly overparametrized NN (1) with $L$ layers and $m$ hidden units per layer and initialization variance $\sigma^2$, where all layers $\theta$ have been trained using gradient descent under squared loss on a consistent dataset $\mathcal{D}$ after random initialization $\theta_0 \sim \mathcal{N}(0,1)$. Then it holds with probability at least $1 - \delta$ over random initialization $\sqrt{L_{\text{test}}^{\text{NN}}} \geq |\sigma^L J(X_{\text{test}}) - O(1/\sqrt{m}) - \sqrt{L_{\text{test}}^{\text{int}}}|$*

For general datasets $\mathcal{D}$, it is highly likely that $J(X_{\text{test}}, \sigma) \neq 0$. To prove the statement we use the decomposition of Lemma 2. Now instead of making the variance small to bound the second term we can use the homogeneity of $\Phi(x, \sigma)$ to show that $J(X_{\text{test}}, \sigma) = \sigma^L J(X_{\text{test}})$. Now using twice the reverse triangle inequality we and the lemma 1 get the desired result. □

Now looking at the expression of Thm. 4 and using again the homogeneity of $\Phi$ we can see that $\sqrt{L_{\text{test}}^{\text{int}}}$ does not change when changing $\sigma$ because the different factors cancel out and the term corresponding to the linear approximation should also not impact the result too much as long as we choose $m$ large and do not increase $\sigma$ significantly. Respecting this constraint we can now easily increase the variance to make the second term as large as we want. Thus, by increasing $\sigma$ we are able to increase the test loss arbitrarily while not changing the training error at all because the second term in our decomposition does not contribute during training time. This shows that without considering an explicit initialization the training error of a NN might tell us very little about the actual test loss. Thus, our results underline the importance of finding good initialization strategies.

## 5 EXPERIMENTS

In this section we give experimental evidence for our theoretical findings: the influence of NN initialization on the generalization performance of fully-trained wide NNs (Sect. 4) as well as the link to interpolating kernel methods in the first place (Sect. 3).

We perform experiments (Fig. 1) in three different settings[3]: (1) A toy experiment, fitting a 2D-dataset with $N = 10$ datapoints sampled uniformly from $f(x) = \exp(-\|x\|^2)$, $x \in [-1, 1]^2$, with a NN of 1 hidden layer ($L = 2$) with $m = 10000$ neurons; here, the "wide NN" requirement $m \gg N$ is satisfied to the highest degree. (2) Fitting $N = 100$ MNIST digits 0 and 1 by a single-hidden-layer network with $m = 4000$ hidden units. In line with our framework and with other works on wide NNs (e.g., (Du et al., 2018b; Arora et al., 2019b)), we fit these classification labels using squared error loss. (3) Fitting MNIST digits 0 and 1 with 2 hidden layers ($L = 3$) of $m = 1500$ neurons each. The reason for using small training sets is to maintain the overparametrization limit $m \gg N$.

In each setting we vary the variance $\sigma^2$ of the NN initialization, performing always 10 repetitions. We stop training at the same "close to zero" training loss for all $\sigma^2$, chosen in each setting so small that empirically the test loss remains almost constant (while the training loss is decreasing further).

The left panels (Fig. 1) show the test losses of the trained NN vs. the linearization $f^{\text{lin}}$ of the initialized NN (Lemma 1), at different NN initialization variances $\sigma^2$. Both curves are very close, confirming the basic premise that the approximation of wide NNs by linear models is quite accurate for our NNs, even though we are not in the guaranteed overparametrized regime (e.g., $m \geq O(N^6)$ from Du et al. (2018b)). The approximation is better for the two MNIST experiments, where we are actually further from the this regime, whereas the larger repetition variances for the 2D bump are due to sampling the training inputs from a small volume of low dimension.

Most strikingly, the test loss of the trained NN depends heavily on its weight initialization $\theta_0$ even though all NNs have been trained to the same low training error. This is shown by the increase at larger initialization variance $\sigma^2$. Via the link to $f^{\text{lin}}$, this can be understood from the test behavior of strongly overparametrized linear models $f^{\text{lin}}$ with big initialization $\theta_0$ (Eq. (14) and Sect. 4.2).

The right panels underline this behavior predicted by our theory quantitatively: The test error of trained ReLU-NNs as well our our analytical lower bound on it (Thm. 4) both behave at big $\sigma^2$ like $\Theta(\sigma^{2L})$ (see also Eq. (12)), with $L$ the number of layers. This in particular shows how one can find natural examples of (overparametrized) NNs, i.e. initialized randomly and trained by gradient descent, that interpolate the training data perfectly, yet with arbitrarily large test error.

In the regime of small $\sigma^2$ on the other hand, the right panels confirm Thm. 3, namely that the test error is close to the (low) test error of the interpolating kernel method $f_\Phi^{\text{int}}$ (Eq. (5)). Note that its test error in the figures is almost constant in $\sigma^2$, as $f^{\text{int}}$ does not depend directly on the weight initialization $\theta_0$ but on the (many and random) NN features $\phi(x)$, which via the kernel $\Phi$ only determine function smoothness. This is a more reasonable modeling behavior than that of wide NNs initialized at large $\sigma^2$.

## 6 DISCUSSION

In this paper we make the bias explicit that is imposed by gradient descent and random initialization on fully-connected strongly overparametrized NNs. We achieve this by decomposing a gradient descent-trained network with (close to) zero training error into an interpolating kernel term and a second term that vanishes on training inputs but not on test inputs.

Because this second term is independent of the training labels but depends strongly on the initialization, it is favorable for good generalization to reduce its impact. This can be achieved by choosing the variance of the NN initialization to be sufficiently small, as we have shown. On the other hand, too small an initialization variance limits the expressivity of the NN and its features.

Our work furthers the practical understanding of modern NNs by utilizing recent theoretical advances in strongly overparametrized NNs. In particular, we saw that the initialized weights are not merely a starting point for training, but significantly influence the performance of the NN trained to

---

[3]More details on the settings needed to reproduce the experiments can be found in Appendix E.

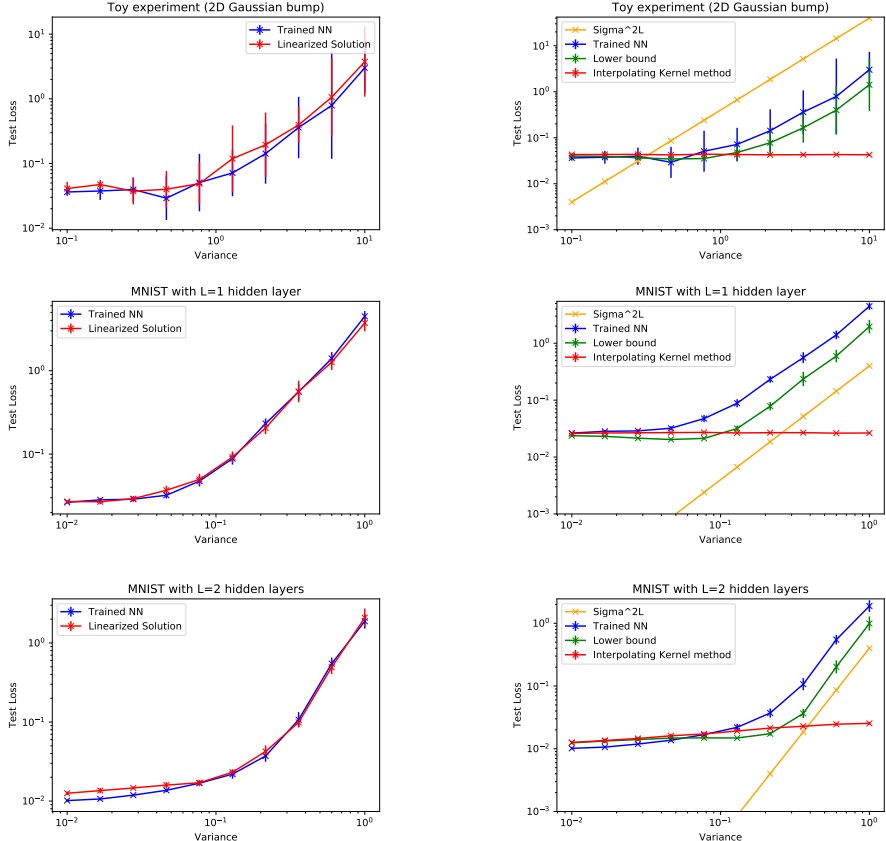

Figure 1: Experiments on the 2D bump $f(x) = \exp(-\|x\|^2)$ (top row) and on MNIST with $L = 2$ and $L = 3$ layers, respectively (last two rows). Left panels: The test error of trained wide NNs as well as their linearized versions increases significantly with the NN initialization variance $\sigma^2$, despite training all NNs to the same very low training error. This is explained by the theory in Sect. 4.2, as the linearized approximation of our NNs is quite accurate (see also Sect. 3). Right panels: Quantitatively, at big $\sigma^2$, the NN test loss as well as its lower bound both grow as $\Theta(\sigma^{2L})$ (Thm. 4). At small $\sigma^2$, on the other hand, the NN test error is close to that of the interpolating kernel method $f_\Phi^{\mathrm{int}}$ (Eq. (5)), confirming Thm. 3.

zero empirical error. Similarly, focussing on the first term of our decomposition, learning-theoretical results on interpolating kernel methods lead to a better understanding of puzzling features of NNs.

Our work provides intuition for the "essentially no barriers" phenomenon first observed in Draxler et al. (2018); Garipov et al. (2018): When scaling all initialization weights $\theta_0$ by a common continuous factor (like $\sigma$), all of the finally trained NNs $f_\sigma^{\mathrm{NN}}$ are connected continuously and their training error stays at (close to) zero. This is because $\Phi$ changes by only a scalar factor $\sigma^{2L}$, which continuously changes the second (test) term in the linearization $f^{\mathrm{NN}} \approx f^{\mathrm{lin}}$ (Eq. 14) but cancels out of the first (training) term. It would be interesting for further work to investigate varying $\theta_0$ more generally in the hypersurface of zero training loss, which induces non-scalar changes in $\Phi$.

The connection of NNs with minimum complexity interpolating kernel methods opens up more directions for future research. An important next step would be to go from strongly overparametrized NNs to merely overparametrized NNs, where the (infinitesimal) kernel dynamics effected by NN training is to be understood in addition to fixed-kernel interpolation. Further, the connection remains to be made explicit for CNNs and RNNs, as well as for other (classification) loss functions, whose strict minimization often requires infinite weights. For SGD at least, instead of full gradient descent, the basic decomposition into a training part and its orthogonal component (Eq. 14) remains valid.

Another direction needed to fully leverage the link to NNs made explicit in Sect. 4.1, is to understand better the properties – e.g. the generalization behavior (Liang & Rakhlin, 2018) – of interpolating kernel methods, whose close connection to large modern-day NNs we have highlighted in this paper.

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

## A APPENDIX

### A MINIMUM 2-NORM SOLUTION

Let us consider an affine linear model $\hat{f}(x) = \Phi(x)\theta$, where $\Phi(x)$ is our feature map and $\theta$ our weights. We want to choose the weights as a solution of the following problem:

$$\tilde{\theta} = \min_{\theta} \frac{\|\theta\|_2^2}{2} \text{ subject to } Y = \Phi(X)\theta$$

First, we use the method of Lagrange multipliers to and define the following Lagrange function

$$\mathcal{L}(\theta, \lambda) = \frac{\|\theta\|_2^2}{2} - \lambda^T \left( \Phi(X)\theta - Y \right)$$

Taking the derivative with respect to $\theta$ and setting it to zero gives us $\tilde{\theta}^T = \lambda^T \Phi(X)$. Now plugging this into the Lagrangian we get

$$\mathcal{L}(\lambda) = -\frac{1}{2}\lambda^T \Phi(X)\Phi(X)^T \lambda + \lambda^T Y$$

Minimizing this term with respect to $\lambda$ we get $\lambda = \left( \Phi(X)\Phi(X)^T \right)^{-1} Y$. Inserting this relation we get

$$\tilde{\theta} = \Phi(X)^T \left( \Phi(X)\Phi(X)^T \right)^{-1} Y$$

### B CONNECTING THE INTERPOLATING MINIMUM 2-NORM SOLUTION TO THE MINIMUM RKHS-NORM INTERPOLATING KERNEL METHOD

Building upon the previous section we now want to show that a linear predictor where the weights are the solution of a minimum 2-norm interpolation with feature map $\Phi(x)$ is equivalent to a minimum complexity interpolating kernel method with

$$\hat{f} = \frac{1}{2} \operatorname*{arg\,min}_{f \in \mathcal{H}_K} \left\| \tilde{f} \right\|_{\mathcal{H}_K} \text{ subject to } Y = f(X),$$

were the kernel of the RKHS is given by $K(x, y) = \langle \Phi(x), \Phi(y) \rangle$ and $\Phi$ is the same function used for the linear ansatz in the minimum 2-norm solution.

We can find the solution of the minimum complexity interpolating kernel method by minimizing the following Lagrangian:

$$\mathcal{L}(f, \lambda) = \frac{1}{2} \left\| \tilde{f} \right\|_{\mathcal{H}_K} + \lambda^T \left( Y - f(X) \right)$$

Because the RKHS-Norm is bounded by definition (for non degenerate data) we can use the representer theorem ($f = \sum_{i=1}^N \alpha_i K(x_i, .)$ with $\alpha_i \in \mathbb{R}$) and the reproducibility of the kernel $K$ to arrive at

$$\mathcal{L}(\alpha, \lambda) = \frac{1}{2}\alpha^T K(X, X)\alpha + \lambda^T \left( Y - K(X, X)\alpha \right)$$

Now taking the derivative with respect to $\alpha$ we get $\alpha = \lambda$. Inserting this into our Lagrangian we get

$$\mathcal{L}(\lambda) = -\frac{1}{2}\lambda^T K(X, X)\lambda + \lambda^T Y$$

This Lagrangian is equivalent to the Lagrangian of the minimal two norm solution in the dual representation and due to the convexity of the problem the solutions must also be equivalent.

### C LEMMA 2

In this section we want to give the proof for lemma 2. Let $f^{NN}$ be a fully-connected NN as defined in the notation section. For simplicity we switch from the NTK initialization to the He et al. (2015) initialization. The proof is equivalent in both parametrizations. The only difference is that we can

suppress some of the scaling factors in the initialization of the weights. We again assume a ReLU-activation function. Before showing the actual lemma we will first show the following expression which we will use to proof the lemma

$$f_W^{NN}(x) = \frac{1}{L}\langle W, \nabla_W f(x)\rangle, \tag{15}$$

were the scalar product is defined as the Hilbert-Schmid-product with $\langle A, B\rangle = tr\left[A^T B\right]$ and $W = diag(W_L, ..., W_1)$ is the diagonal matrix with all the weigh matrices on the diagonal. We restrict ourselves to networks without biases but the proof with biases is almost identical.

To prove the lemma we can first proof that $\forall l \in \{1, ..., L\}$ we have $f_{W_l}^{NN}(x) = \langle W_l, \nabla_{W_l} f(x)\rangle$. If this statement is true it is easy to see that the formula above is also true.

A ReLU Network can be written in the following way

$$f_W^{NN}(x) = W_L \mathbf{l}_{h^{L-1}(x)\geq 0} W_{L-1}...\mathbf{l}_{h^1(x)\geq 0} W_1 x, \tag{16}$$

where $\mathbf{l}_{h^l(x)\geq 0}$ is the diagonal matrix with the step function on the diagonal corresponding to the components of $h^l(x) \geq 0$. Now, we can define $a(W_{L:l}, x) := W_L \mathbf{l}_{h^{L-1}(x)\geq 0} W_{L-1}...\mathbf{l}_{h^l(x)\geq 0}$ and $b(W_{l:1}, x) := \mathbf{l}_{h^{l-1}(x)\geq 0} W_{l-1}...$ to get

$$f_W^{NN}(x) = a(W_{L:l}, x) W_l b(W_{l:1}, x)$$

Using the fact that $\frac{\partial a_{ij}}{\partial (W_l)_{mn}} = 0$ because we define $\frac{\partial (\mathbf{I}_{h^l(x)=0})_{ij}}{\partial (W_l)_{mn}} = 0$. Defining the derivative of the stepfunction at the step to be zero is also done in most of the standard frameworks like tensorflow. Additionally, the region where the stepfunction jumps is a $d - 1$ dimensional subspace and thus the expression holds almost everywhere anyway. Now, since our weights do not move much and only a neglectable number change the step function we can just define this derivative to be zero. The derivative of $b(W_{l:1}, x)$ with respect to $W_l$ is clearly zero because it does not depend on $W_l$.

Using this we can see that almost everywhere

$$\langle W_l, \nabla_{W_l} f^{NN}(x)\rangle = \sum_{nm}(W_l)_{nm}\frac{\partial f_W^{NN}(x)}{\partial (W_l)_{nm}} = a(W_{L:l}, x) W_l b(W_{l:1}, x) = f^{NN}(x),$$

Applying this result $L$-times we have get that almost everywhere $f^{NN}(x) = \frac{1}{L}\langle W, \nabla_W f(x)\rangle$. Thus, we get almost everywhere

$$f_{\theta_0}(x) = \frac{1}{L}\langle \theta_0, \Phi(x)\rangle$$

and therefore also

$$f_{\theta_0}^{NN}(x) - \Phi(x)\Phi(X)^T(\Phi(X)\Phi(X)^T)^{-1}f_{\theta_0}^{NN}(X) = \frac{1}{L}\Phi(x)\left(\mathbf{1} - \Phi(X)^T(\Phi(X)\Phi(X)^T)^{-1}\Phi(X)\right)\theta_0$$

## D    Lemma 3

In this section we want to show the statement of lemma 3. We want to show that

$$\mathbb{P}\left(\frac{1}{L}|\Phi(x)\left(\mathbb{I} - \Phi(X)^T(\Phi(X)\Phi(X)^T)^{-1}\Phi(X)\right)\theta_0| < \sigma^{2L}\delta\right) \geq 1 - \delta$$

First, let us focus on the operator $\left(\mathbb{I} - \Phi(X)^T(\Phi(X)\Phi(X)^T)^{-1}\Phi(X)\right)$. The second term has the properties of a projection operator (symmetric and equal to its square). Thus, all the eigenvalues are either zero or one. The operator projects $M$-dimensional vectors into a $N$-dimensional subspace and then back to the $M$-dimensional space. Thus, it has $N$ eigenvalues which are one and all the rest are zero. Now, looking at the whole operator we see that it has $M - N$ eigenvalues which are one and the rest are zero. Therefore, we can neglect the projection operator because this will only make our result larger due to the neglected $N$ dimensions which would be projected out.

Using this and the formula we have derived in th proof for lemma 2 we get almost everywhere

$$\frac{1}{L}|\Phi(x)\left(\mathbb{I} - \Phi(X)^T(\Phi(X)\Phi(X)^T)^{-1}\Phi(X)\right)\theta_0| \leq |\frac{1}{L}\Phi(x)\theta_0| = |f_{\theta_0}^{NN}(x)|$$

Now, we can use Markov's inequality to arrive at

$$\mathbb{P}\left(\left|\frac{1}{L}\Phi(x)\left(\mathbb{I}-\Phi(X)^T(\Phi(X)\Phi(X)^T)^{-1}\Phi(X)\right)\theta_0\right|^2 < \mathbb{E}\left[|f_{\theta_0}^{\text{NN}}(x)|^2\right]/\delta\right) \geq 1-\delta$$

Thus, the only thing that is left to do is to show that $\mathbb{E}\left[f_{\theta_0}^{\text{NN}}(x)\right] \leq \sigma^{2L}$. To start of we can again write the Neural Network as a product of matrices

$$f_{\theta_0}^{\text{NN}}(x) = W_L \mathbf{l}_{h^{L-1}(x)\geq 0}W_L...\mathbf{l}_{h^0(x)\geq 0}W_1 x.$$

We use a similar approach as Du & Hu (2019) and generalize this approach to non-linear networks. We make use of the fact that for for a random matrix $A \in \mathbb{R}^{d_1 \times d_2}$ with i.i.d. $\mathcal{N}(0,1)$ entries and a arbitrary non-zero vector $v \in \mathbb{R}^{d_1}/0$ the distribution of $\frac{\|Av\|^2}{\|v\|^2}$ is $\chi^2_{d_2}$ distributed. Additionally, it is easy to see that $\frac{\|\mathbf{1}_{Av\geq 0}Av\|^2}{\|Av\|^2} \leq 1$. Defining $Z_i = \frac{\|W_i\mathbf{l}_{h^{i-1}(x)\geq 0}W_{i-1}...\mathbf{l}_{h^1(x)\geq 0}W_1 x\|^2}{\|\mathbf{l}_{h^{i-1}(x)\geq 0}W_i...\mathbf{l}_{h^1(x)\geq 0}W_1 x\|^2}$ we thus get

$$\mathbb{E}\left[|f_{\theta_0}^{\text{NN}}(x)|^2\right] \leq \mathbb{E}[Z_1 \ldots Z_L] = \Pi_{l=1}^L \mathbb{E}[Z_l] = \sigma^{2L}.$$

Inserting the result for $\mathbb{E}\left[f_0(x,\theta_0)\right]$ we get

$$\mathbb{P}\left(\frac{1}{L}|\Phi(x)\left(\mathbb{I}-\Phi(X)^T(\Phi(X)\Phi(X)^T)^{-1}\Phi(X)\right)\theta_0| < \sigma^L/\sqrt{\delta}\right) \geq 1-\delta.$$

## E   DETAILS ON OUR EXPERIMENTS

In this section we want to talk about how we conducted the experiments. Since the experimental settings are not very complicated this should be more than enough to easily reproduce our results.

As described in the experimental section we used three different setting to investigate the dependence of the Neural Network solution on the initialization variance. We used a standard He et al. (2015)-initialization which is, as mention in the notation section, equivalent to the NTK-parametrization and standard normal initialization when we rescale the learning rate. Since we choose the theoretical value of $\eta = \frac{1.9}{\lambda_{max}(\Psi(X)\Psi(X)^T)}$ for the learning rate the settings are equivalent. For all experiments we use Tensorflow with ReLU-activation functions, set the bias initially to zero and train with GD. For all networks we use a low number of training data points to make sure that we are in or close to the over-parametrized regime. This is necessary to assure the validity of the linearisation.

The first Network for the 2D exponential Bump is a single hidden layer NN with 10000 hidden units. We use 10 training data points and 300 testing points. We train 10 networks for each variance value and until the training loss drops first bellow a loss of $10^{-5}$. The final training loss of the curves plotted in the experimental section can be seen in figure 2. We choose the stopping point for the training error in such a way that even though the training loss is still decreasing further the test loss is almost constant.

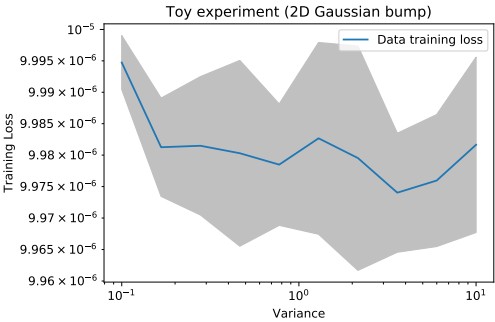

Figure 2: Benchmark experiment with a 2D-Gaussian Bump

For the single layer MNIST network we use 4000 hidden units and for the two layer network two layers of each a 1500 hidden units. We use 100 training samples of 0 and 1 labelled examples of the

training set and 100 samples from the test set to calculate the test error. For the MNIST Network we train the networks to reach a training error smaller than $10^{-4}$. The plot of the final training errors that corresponds to the curves in the Experimental section are shown in figure 3.

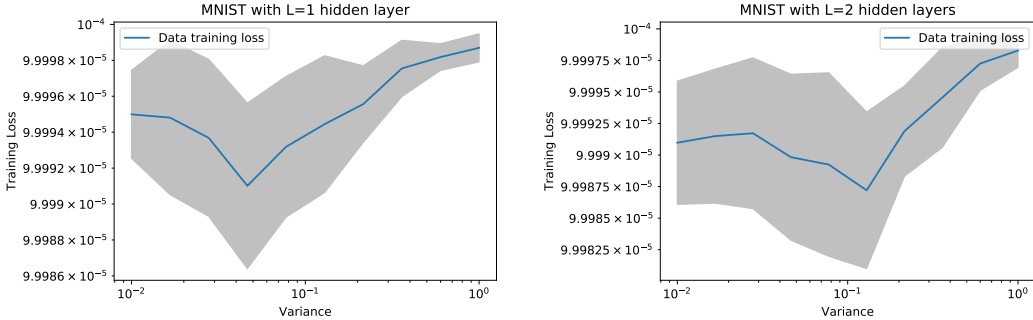

Figure 3: Experiments with MNIST for a NN with 1 and 2 hidden layers

