# OpenReview forum: "Wide Neural Networks are Interpolating Kernel Methods: Impact of Initialization on Generalization"
_ICLR.cc/2020/Conference — Reject_

### Official Review · AnonReviewer1 · 2019-10-23
**Official Blind Review #1**

**Rating:** 1

**Review:**


[Summary]
This paper studies the impact of initialization noise on the theories of wide neural networks in the Neural Tangent Kernels (NTK) regime. The paper proves that the difference between the trained neural net and the kernel interpolator (with the NTK) can be bounded by O(\sigma^L + 1/\sqrt{m}), where \sigma^2 is the initializing variance of each individual weight entry. Relationships between the generalization error of these two functions are derived from the above bound.

[Pros]
The general message that this paper conveys is interesting -- the initial network f_{\theta_0}(x), which is typically omitted (or made small by making \sigma small) in NTK analyses, can deviate the converged NN from the kernel interpolator in terms of generalization error.

[Cons]
There are fundamental mistakes in the statements/proofs of Theorem 2, 3, 4:
-- Theorem 2: the statement is “whp over W, the bound … holds uniformly for x”. The proof relies on Lemma 3, whose statement is also uniform over x, but the proof applies the Markov inequality *for a single x* and is thus valid only for a single x. (As it’s Markov, it seems not sensible to apply the union bound upon it.)

-- Theorem 3: the difference between L^NN_test and L^int_test should be on the order of (\sigma^L + 1/\sqrt{m}) rather than it squared. To bound the difference in squared loss we have a^2 - b^2 <= O(1) * |a-b| (if a, b are bounded by O(1)). We don’t have a^2 - b^2 <= |a - b|^2.

-- Theorem 4: J(X_test) as defined is a vector whose dimension grows with the number of test data points, where the theorem requires it to be a scalar. Indeed the treatment of test data as a fixed matrix (rather than samples from a distribution) is already a bit atypical.

***

I have read the authors' rebuttal and the other reviews, and I'm glad to see the issues with Theorem 3 and 4 pointed out above are fixed in the revision. However, I also agree with the other reviewers that the paper in the present stage has not yet demonstrated sufficient technical contributions, and thus I am keeping my original evaluation.

**Experience Assessment:**

I have published one or two papers in this area.

**Review Assessment: Checking Correctness Of Derivations And Theory:**

I carefully checked the derivations and theory.

**Review Assessment: Checking Correctness Of Experiments:**

N/A

**Review Assessment: Thoroughness In Paper Reading:**

I read the paper at least twice and used my best judgement in assessing the paper.

---

> ### Author Response · Authors · 2019-11-15
> **Re: Official Blind Review #1**
>
> We thank reviewer 1 for their careful review, in particular for pointing out mistakes in our theorems. We appreciate that the reviewer finds our main result interesting.
>
> We acknowledge the errors pointed out by the referee, but believe that these do not alter the main message of our paper significantly. We address the reviewer comments concerning our theorems in turn:
>
> -- Theorem 2: We agree and changed the order of quantifiers in the theorem. Despite this mistake, Thm. 3 remains valid as one does not need the union bound to derive it from Lemma 3; instead, one can directly apply Markov's inequality to || f_int(X_test) –f_lin(X_test)||^2/N_{test} rather than to || f_int(x) –f_lin(x)||^2 as done in the proof of Lemma 3. We describe this proof strategy now below Thm. 3.
>
> -- Theorem 3: We agree with the referee that there was a mistake in the statement of the inequality. Instead the bound should have been:
> L^NN_test <= (sqrt(L^int_test)+O(1/sqrt(m))+sigma^L/sqrt(delta))^2,
> or equivalently, as we write in the updated version:
> sqrt(L^NN_test) <= sqrt(L^int_test)+O(1/sqrt(m))+sigma^L/sqrt(delta).
>
> -- Theorem 4: We mistakenly omitted a 2-norm-sign in the definition of J(X_test,sigma), which makes this quantity a scalar. We have corrected this.

---

### Official Review · AnonReviewer3 · 2019-10-23
**Official Blind Review #3**

**Rating:** 6

**Review:**

The paper considers the impact of initialization bias on test error in strongly overparameterized neural networks. The study uses tools from recent literature on the generalization of overparameterized neural networks, i.e. neural tangent kernels and interpolating kernel method, to provide useful insights on how the variance of weights initialization affects the test error. I have a few questions about theoretical results, but the paper has a convincing experiment that supports its theoretical claims. Addressing the following points will improve the exposition of the paper.
1. Please provide a little hint on how Lemma 2 rewrites the equation (13) for linearized function for easier readability without referring to the Appendix.
2. In the case of cross-entropy error, would the effect be similar? Could this be verified with a similar experiment as for MSE?
3. To what extent this result is observed in not as strongly overparameterized settings? In other words, it would be interesting to see what happens if you fix the architectural choice while increasing the number of training parameters, how long does the test error effect persist?

Minor remark:
- a few typos are present on pages 4, 5, 7, 8

**Experience Assessment:**

I have read many papers in this area.

**Review Assessment: Checking Correctness Of Derivations And Theory:**

I assessed the sensibility of the derivations and theory.

**Review Assessment: Checking Correctness Of Experiments:**

I assessed the sensibility of the experiments.

**Review Assessment: Thoroughness In Paper Reading:**

I read the paper at least twice and used my best judgement in assessing the paper.

---

> ### Author Response · Authors · 2019-11-15
> **Re: Official Blind Review #3**
>
> We thank reviewer 3 for their review and positive assessment of our work.
>
> 1. In the updated version we have added before Lemma 2 the main identity used for proving it. Applying this identity twice in Eq. (13) leads to Eq. (14).
>
> 2. A fundamental difference between MSE and cross-entropy loss is that the minimum of the cross-entropy is not attained at finite weight values and training becomes exponentially slow. For cross-entropy, the linearized NN can be investigated numerically but there is no closed-form solution for the training behavior available (as Eq. (14) for MSE). Because no such decomposition is available, the same experiments as in our paper cannot be done for cross-entropy loss; but one could still experimentally investigate the behavior of the test error with respect to changing the variance.
>
> 3. We agree with the referee that it would be interesting to investigate how the effect changes when the number of parameters is being varied.
>
> We have fixed typos and harmonized notation in several parts of the paper.

---

### Official Review · AnonReviewer2 · 2019-10-27
**Official Blind Review #2**

**Rating:** 1

**Review:**

This paper studies the solution of neural network training in the NTK regime. The trained network can be written as the sum of two terms --- the first is the minimum RKHS norm interpolating solution, and the second term depends on the initialization. When the initialization scale is small, the second term almost vanishes, but when the initialization scale is large, it's likely that the second term becomes very large, leading to worse generalization.

The technical contribution of this paper is pretty low. The most important formula is (14), which only appears in the second half of the paper (the first half of the paper is almost all known results). The bounds in later part of the paper are also straightforward. Moreover, another paper https://arxiv.org/abs/1905.07777 already studied the same question and showed that non-zero output can increase the generalization error.


-----------
update:
I have read the authors' response. My assessment stays the same since I still think that the technical contribution of this paper is quite limited.

Also there is a negative effect of using small init, which the authors might have overlooked: when the init is smaller, you'd need a larger width for the NN to be in the NTK regime. See e.g. "Fine-Grained Analysis of Optimization and Generalization for Overparameterized Two-Layer Neural Networks. Sanjeev Arora, Simon S. Du, Wei Hu, Zhiyuan Li, Ruosong Wang. ICML 2019".

**Experience Assessment:**

I have published one or two papers in this area.

**Review Assessment: Checking Correctness Of Derivations And Theory:**

I assessed the sensibility of the derivations and theory.

**Review Assessment: Checking Correctness Of Experiments:**

I assessed the sensibility of the experiments.

**Review Assessment: Thoroughness In Paper Reading:**

I read the paper at least twice and used my best judgement in assessing the paper.

---

> ### Author Response · Authors · 2019-11-15
> **Re: Official Blind Review #2**
>
> We thank reviewer 2 for their review and in particular for making us aware of the closely related preprint arXiv:1905.07777.
>
> It is true that our Eq. (14) is the main step from which we derive all our insights on the initialization of NNs, and that this equation could be obtained by specializing Thm. 2 of 1905.07777 to ReLU networks. In order to get rid of the deviation from the kernel interpolator, the preprint 1905.07777 suggests to extend the NN asymmetrically to twice the size (ASI trick), which doubles training time; whereas our work suggests to initialize the NN at small variance (Thm. 2 and 3) to get rid of this term (not exactly but to a high degree ~\sigma^L; as we show, small \sigma>0 has no negative impact on expressivity or training behavior).
>
> Furthermore, our work makes the effect of large initialization on the test error quantitatively explicit (Thm. 4), with a \sigma^L behavior.
>
> We have added and discussed the reference.

---

### Official Review · AnonReviewer4 · 2019-10-31
**Official Blind Review #4**

**Rating:** 3

**Review:**

This paper studies overparameterized fully-connected neural networks trained with squared loss. The authors show that the resulting network can be decomposed as a sum of the solution of a certain interpolating kernel regression and a term that only depends on initialization. Based on this, the authors also derive a generalization bound of deep neural networks by transferring it to a kernel method. My major concern about this paper is the novelty and significance of its results:

In terms of connection to NTK, It seems that the connection between neural networks trained with squared loss and the result of NTK-based kernel regression has already been well-studied by

Arora, Sanjeev, Simon S. Du, Wei Hu, Zhiyuan Li, Ruslan Salakhutdinov, and Ruosong Wang. "On exact computation with an infinitely wide neural net." arXiv preprint arXiv:1904.11955 (2019).

which is a missed citation. Without a clear explanation on the difference between the submission and this paper above, I don’t think this paper is ready for publication.

In terms of generalization, it is also very difficult to judge whether this paper's result is novel. In fact this paper misses almost all citations on generalization bounds for neural networks. Moreover, the generalization bound given in this paper does not seem to be very complete and significant, since the authors do not show when can L_{test}^{int} be small. To demonstrate the novelty and significance of the result, the authors should at least compare their generalization result with the following generalization bounds for over-parameterized neural networks in Section 4:

Allen-Zhu, Zeyuan, Yuanzhi Li, and Yingyu Liang. "Learning and generalization in overparameterized neural networks, going beyond two layers." arXiv preprint arXiv:1811.04918 (2018).
Cao, Yuan, and Quanquan Gu. "A generalization theory of gradient descent for learning over-parameterized deep relu networks." arXiv preprint arXiv:1902.01384 (2019).
Arora, Sanjeev, Simon S. Du, Wei Hu, Zhiyuan Li, and Ruosong Wang. "Fine-grained analysis of optimization and generalization for overparameterized two-layer neural networks." arXiv preprint arXiv:1901.08584 (2019).
Cao, Yuan, and Quanquan Gu. "Generalization Bounds of Stochastic Gradient Descent for Wide and Deep Neural Networks." arXiv preprint arXiv:1905.13210 (2019).

Overall, I suggest that the authors should make a clear discussion on the relation of this paper to many existing works mentioned above. As long as the authors can give a convincing demonstration of the novelty and significance of their results, I will be happy to increase my score.

A minor comment: how can the bound in Theorem 3 be derived based on Theorem 2? Should there be a constant factor in the bound?



**Experience Assessment:**

I have published one or two papers in this area.

**Review Assessment: Checking Correctness Of Derivations And Theory:**

I assessed the sensibility of the derivations and theory.

**Review Assessment: Checking Correctness Of Experiments:**

I assessed the sensibility of the experiments.

**Review Assessment: Thoroughness In Paper Reading:**

I read the paper at least twice and used my best judgement in assessing the paper.

---

> ### Author Response · Authors · 2019-11-15
> **Re: Official Blind Review #4**
>
> We thank reviewer 4 for their review.
>
> We should have cited and discussed the paper arXiv:1904.11955 and have added the citation now. This paper in effect discusses only the interpolating first term of our decomposition in Eq. (14), whereas our work focuses on the presence and effect of the initialization-dependent second term of this decomposition, which is responsible for all our results. This second term does not appear in 1904.11955, as these authors set the initial NN output to 0 by fiat. This second term is significant as shown by our Thm. 4 and confirmed by the experiments.
>
> We acknowledge that more work has to be done to compare existing NN generalization bounds with bounds obtained from our Thm. 3. We see a main contribution of our work in establishing this link between generalization bounds for kernel methods and for NNs, and show in the experiments that this link is tight (for small initialization sigma). The fact that the interpolating kernel term often generalizes well is shown experimentally in 1904.11955, and theoretically analyzed in [Liang,Rakhlin 2018].
>
> Theorem 3 can be proven without applying the union bound, thus without a constant factor. See the added comment below Theorem 3 and our reply to reviewer 1.

---

### Public Comment · ~Difan_Zou1 · 2019-10-24
**Interesting work**

It is a very interesting and important finding that the initialization scheme has a significant impact on the generalization performance of neural networks trained by gradient descent.

I would like to point out a few papers on the theoretical understanding of over-parameterized deep neural networks, which are also very related to your work

[1] Amit Daniely. SGD learns the conjugate kernel class of the network. In Advances in Neural Information Processing Systems, pages 2422–2430, 201.
[2] Difan Zou, Yuan Cao, Dongruo Zhou, and Quanquan Gu. Stochastic gradient descent optimizes over-parameterized deep relu networks. arXiv preprint arXiv:1811.08888, 2018.
[3] Yuan Cao and Quanquan Gu. A generalization theory of gradient descent for learning overparameterized deep ReLU networks. arXiv preprint arXiv:1902.01384, 2019.

---

> ### Author Response · Authors · 2019-10-28
> **Re: Interesting work**
>
> Dear Difan Zou,
> thank you for your interest and positive feedback. We were very happy to learn about these relevant references.
>
> Ref. [1] by Daniely is interesting as it is a very early paper using the connection of NNs with kernel methods to investigate the convergence of SGD and is a precursor to the works by Du et al. (2018b) and Li & Liang (2018), which our paper builds on.
>
> Ref. [2] by Zou et al. has similar relevance to our paper as the works by Du et al. (2018a) and Allen-Zhu (2018a) have, generalizing (S)GD convergence results to deep NNs with a focus on a variety of loss functions. Besides giving connections kernel methods, the (polynomial-time) convergence guarantees of all mentioned papers motivate why the NNs in our main theorems can be assumed to be fully-converged.
>
> Ref. [3] by Cao and Gu is not directly connected to our work. It investigates generalization in NNs per se, whereas our paper connects generalization of NNs with the generalization in the associated kernel method.
>
> We will definitely cite Refs. [1,2] in the next version of our paper.
>
> Many thanks,
> The authors

---

### Decision · Program_Chairs · 2019-12-19

**Decision:**

Reject

**Comment:**

This paper proves that fully-connected wide ReLU-NNs trained with squared loss can be decomposed into two parts: (1) the minimum complexity solution of an interpolating kernel method, and (2) a term depends heavily on the initialization. The main concerns of the reviewers include (1) the contribution are not significant at all given prior work; (2) flawed proof,  and (3) lack the comparison with prior work. Even the authors addressed some of the concerns in the revision, it still does not gather sufficient support from the reviewers after author response. Thus I recommend reject.